# Innovative Polarimetric Interferometric Synthetic Aperture Radar Land Cover Classification: Integrating Power, Polarimetric, and Interferometric Information for Higher Accuracy

**DOI:** 10.3390/s25071996

**Published:** 2025-03-22

**Authors:** Yifan Xu, Aifang Liu, Youquan Lin, Moqian Wang, Long Huang, Zuzhen Huang

**Affiliations:** Nanjing Research Institute of Electronics Technology, Nanjing 210039, China; njuxyf@126.com (Y.X.); linyouquan1965@163.com (Y.L.); wangmoqian2000@163.com (M.W.); imhuanglong@163.com (L.H.); hzzhit@126.com (Z.H.)

**Keywords:** PolInSAR, land cover classification, information fusion, vegetation classification

## Abstract

The Polarimetric Interferometric Synthetic Aperture Radar (PolInSAR) system is a combination of polarimetric SAR and interferometric SAR, which can simultaneously obtain the power information, polarimetric information, and interferometric information of land cover. Traditional land cover classification methods fail to fully utilize these information types, resulting in limited classification types and low accuracy. This paper proposes a PolInSAR land cover classification method that fuses power information, polarimetric information, and interferometric information, aiming to enrich the classification types and improve the classification accuracy. Firstly, the land cover is divided into strong scattering areas and weak scattering areas by using the power information to avoid the influence of weak scattering areas on the classification results. Then, the weak scattering areas are distinguished into shadows and water bodies by combining the interferometric information and image corners. For the strong scattering areas, the polarimetric information is utilized to distinguish vegetation, buildings, and bare soil. For the vegetation area, the concept of vegetation ground elevation is put forward. By combining with the anisotropy parameter, the vegetation is further subdivided into tall coniferous vegetation, short coniferous vegetation, tall broad-leaved vegetation, and short broad-leaved vegetation. The effectiveness of the method has been verified by the PolInSAR data obtained from the N-SAR system developed by Nanjing Research Institute of Electronics Technology. The overall classification accuracy reaches 90.2%, and the Kappa coefficient is 0.876.

## 1. Introduction

Polarimetric Interferometric SAR (PolInSAR), an advanced technology that combines polarimetric SAR and interferometric SAR, not only reveals the polarimetric characteristics of ground objects, providing conditions for in-depth exploration of properties of ground objects, but it can also accurately capture the interferometric information of ground objects, thereby making precise measurements of the elevation of ground objects. Therefore, in-depth analysis of PolInSAR data for land cover classification has shown great application potential and theoretical value in multiple fields [1,2,3,4,5,6], including military, forestry, agriculture, and more.

However, how to efficiently integrate power information, polarimetric information, and interferometric information in PolInSAR to improve the categories and accuracy of land cover classification remains a technical problem that urgently needs to be solved currently. The previous methods of PolInSAR land cover classification [1,2,3,4,5,6] used these three types of information in a chaotic or insufficient way. Especially for power information, most of them did not distinguish between strong scattering areas and weak scattering areas, which would lead to a large number of misclassifications.

Verma et al. [1] proposed some classification frameworks based on deep learning for processing PolInSAR data. These frameworks are capable of automatically learning the complex patterns in data, thereby improving the accuracy of land cover classification.

Kumar et al. [2] adopted the following approach for land cover classification: Firstly, preprocess spaceborne polarimetric SAR data and hyperspectral data through operations such as normalization and registration to prepare them for fusion. Secondly, utilize specific fusion algorithms (which differ based on the particular research) to integrate the advantages of the polarization characteristics of polarimetric SAR and the high spectral resolution of hyperspectral data. Lastly, input the fused data into selected classification models, like decision trees or neural networks, and configure the parameters for the training set and validation set to accomplish the land cover classification task.

Diniz et al. [3] employed two different machine learning classifiers, namely, Random Forest (RF) and support vector machine (SVM), and utilized different scenarios that combined backscattering, polarimetry, and interferometry for classification. The classification process was divided into two stages to enhance the accuracy of the results.

These methods have achieved good classification results in specific PolInSAR data, but their robustness remains to be verified.

Power, as the key information of PolInSAR, directly reflects the scattering intensity of ground objects to electromagnetic waves. Based on the difference in power, we can distinguish between weak scattering areas and strong scattering areas. Due to the extremely low power in the weak scattering area, it is almost submerged in a system’s thermal noise, making polarimetric information and interferometric information basically ineffective in this area. Therefore, special treatment needs to be carried out on the weak scattering area.

The strong scattering area contains rich and clear polarimetric information and interferometric information, providing strong support for land cover classification. In fact, numerous references [1,2,3,4,5,6] have already successfully utilized this information to conduct detailed classifications of land cover. Usually, in the classification process, polarimetric information is first extracted through polarimetric decomposition techniques, and then an appropriate classifier is selected for precise classification to accurately assign each pixel to the corresponding land cover category.

Interferometric information also plays an irreplaceable role in land cover classification, especially demonstrating its unique advantages when distinguishing between high and low ground objects. Through interferometric information, we can obtain the DEM (Digital Elevation Model) of a ground object’s surface, that is, the absolute elevation of the ground object relative to a certain reference plane. However, it should be noted that the DEM itself is not sufficient to directly support land cover classification. To classify more accurately, we need to further extract the ground elevation of the ground object, that is, the height difference in the ground object relative to the surrounding ground. This ground elevation can more accurately reflect the height characteristics of the ground object itself.

Taking the vegetation area as an example, we first use polarimetric information to conduct a preliminary classification of the area to identify the vegetation and soil areas. Then, by using the obtained DEM data of the vegetation area, we further extract the vegetation ground elevation. In this way, the high vegetation area and the low vegetation area are clearly distinguished. Similarly, for other types of land cover, we can also extract the ground elevation information through similar methods.

Neumann et al. [7] first proposed the anisotropic parameter, which can characterize the transformation of scattering particles from spherical to dipolar within the polarization plane. It was found that at X-band frequencies, the wavelength is comparable to the size of vegetation leaves, so the anisotropic parameter is used to distinguish different leaf types of vegetation and further classify the vegetation into coniferous and broad-leaved vegetation.

Given the limitations of the existing PolInSAR land cover classification methods, this paper innovatively proposes a land cover classification method that can comprehensively utilize power, polarimetric, and interferometric information. This method not only improves the classification categories but also has high accuracy and reliability, providing strong support for the application of PolInSAR technology.

## 2. Research Methodology and Framework

This paper will describe in detail the steps of the proposed method. Figure 1 shows the overall framework of the method, revealing how to fuse the power information, polarimetric information, and interferometric information in PolInSAR data for land cover classification.

Specifically, the overall framework of the method in this paper includes data preprocessing, acquisition of the DEM, segmentation of the polarimetric SAR image, division of strong and weak scattering areas, classification of weak scattering areas, classification of strong scattering areas, and classification of vegetation.

Using spaceborne or airborne PolInSAR systems to acquire PolInSAR data, and through RD (Range-Doppler), CS (Chirp Scaling), or other imaging processing techniques, we obtained eight complex images, namely, four primary images, represented by HH, HV, VH, and VV, and four secondary images, represented by $HH^{\prime }$, $HV^{\prime }$, $VH^{\prime },$ and $VV^{\prime }$.

Firstly, we preprocessed these data, including operations such as radiometric calibration [8], polarimetric interferometric calibration [9], and speckle filtering [10], with the aim of ensuring the correctness of the amplitude and phase of the data and reducing the impact of speckle noise.

In the process [11] of acquiring the DEM, we utilized the complex images to acquire the elevation of the target area. This process mainly includes steps such as image registration, removal of the flat-earth phase, phase unwrapping, and phase filtering.

For the segmentation of polarimetric SAR images, we fused the information of multiple polarimetric channels to divide the image into multiple areas with similar properties. These operations, as conventional means of PolInSAR data processing, are not the focus of this paper. Readers can refer to the literature [12] in relevant fields for a more in-depth understanding.

We classified land cover into strong scattering areas and weak scattering areas based on the power information. The relevant content will be elaborated in detail in Section 2.1. We classified the weak scattering areas by combining the DEM and the characteristics of image corners. The specific details will be expounded in Section 2.2. We classified the strong scattering areas into bare soil, buildings, and vegetation through polarimetric decomposition. The relevant content will be thoroughly discussed in Section 2.3.

For the vegetation extracted from the strong scattering areas, we introduced the concept of ground elevation and combined it with the anisotropy parameter to achieve a refined classification of the vegetation, subdividing it into tall coniferous vegetation, short coniferous vegetation, tall broad-leaved vegetation, and short broad-leaved vegetation. This part of the content will be introduced in Section 2.4.

After the data preprocessing, DEM acquisition, and polarimetric SAR image segmentation, we not only obtained height information for each pixel, but we also clustered homogenous pixels into the same grid. The subsequent classification process assumes that the data preprocessing, DEM acquisition, and polarimetric SAR image segmentation have been completed.

### 2.1. Distinguish Strong Scattering Areas and Weak Scattering Areas by Using Power Information

In the PolInSAR system, the backscattering intensities of different ground objects vary, while the system thermal noise remains constant. This leads to differences in the signal-to-noise ratios of different ground objects in the same image. The sensitivity of the PolInSAR system is usually measured by NESZ (Noise Equivalent Sigma Zero), as shown in Equation (1):(1)$$NESZ=\frac{2{\left(4\pi \right)}^{3}R^{3}F_{n}kTV_{st}L_{s}}{P_{av}\lambda^{3}G^{2}\delta_{rg}}$$
where R represents the slant range between the target and the radar; $F_{n}$ is the system noise figure; k is the Boltzmann constant; T is the absolute temperature of the system; $V_{st}$ is the relative velocity between the radar and the target; $L_{s}$ is the system loss; $P_{av}$ is the average transmitted power of the system; λ is the working wavelength of the radar; G is the antenna gain; and $\delta_{rg}$ is the ground range resolution.

NESZ defines the minimum backscattering coefficient of distributed targets that can be accurately detected by the PolInSAR system. When the backscattering coefficient is higher than NESZ by a certain number of decibels, the signal-to-noise ratio of the final output image of the radar system will also be increased by the same number of decibels correspondingly. Specifically, when the signal-to-noise ratio is 0 dB, that is, when the signal is equal to the noise, the average backscattering coefficient of the distributed targets that the system can detect reaches the minimum value, which is NESZ.

In actual SAR images, the areas where the backscattering coefficient is close to or lower than NESZ are referred to as weak scattering areas. The scattering power in these areas is extremely low, and the signal-to-noise ratio is close to or lower than 0 dB, causing the polarimetric information and interferometric information to be submerged by the thermal noise of the radar system, making it impossible to accurately extract them for classification. Therefore, in the actual process of land cover classification, it is necessary to distinguish between weak scattering areas and strong scattering areas and adopt different classification strategies for different areas.

Suppose that the data preprocessing, DEM acquisition and polarimetric SAR image segmentation have been completed. We have not only obtained the height information of each pixel, but also clustered the homogeneous pixels into the grid Gd. The following presents the division method of strong and weak scattering areas.

Based on the segmentation results of the polarimetric SAR image and the power information of the complex images, calculate the average power of the pixels within each segmented grid. Suppose there are n pixels within the grid Gd; then, the average power of the pixels within the grid Gd is calculated by Equation (2):
(2)$$SPAN_{G}=\frac{\sum_{(i,j)\in G}\left[HH^{2}\left(i,j\right)+HV^{2}\left(i,j\right)+VH^{2}\left(i,j\right)+VV^{2}\left(i,j\right)\right]}{4n}$$ where HH(i,j), HH(i,j), HH(i,j), and HH(i,j) represent the amplitudes of pixels at coordinates (i,j) in the four complex images obtained by the main antenna, and the coordinates (i,j) are restricted within the grid Gd.Select a grid that is known to be a shadow area as a reference, and set its average power multiplied by M as the threshold $T_{s}$. To account for tolerance, the value of M can be selected between 1 and 2, which is flexibly chosen based on the actual classification effect. Then, based on the comparison of the average power of pixels within the grid with the threshold $T_{s}$, areas with an average power less than or equal to the threshold $T_{s}$ are classified as weak scattering areas, while areas with an average power greater than the threshold $T_{s}$ are classified as strong scattering areas.

Through the above steps, we can effectively distinguish strong and weak scattering areas in SAR images, providing an important basis for subsequent classification processing.

### 2.2. Classification of Weak Scattering Areas

For SAR data, weak scattering areas are primarily composed of shadows and water bodies. Shadow areas form when objects block electromagnetic waves, preventing specific areas from being illuminated by the electromagnetic waves. Inland water bodies, with small waves, have electromagnetic waves that approximate mirror reflection on their surface. When the radar has a large incidence angle with respect to the observation area, the energy returned to the radar is relatively weak, resulting in weaker echo power in the water body areas.

Based on the radar’s geometric observation of the ground, a model can be established to achieve the classification of shadows and water bodies. Figure 2 shows the geometric observation diagram of shadow formation in SAR images. The observation radar is located at point A; the radar incident angle is *θ*. Suppose there is a ground object at point B with a height H relative to the ground. Since the electromagnetic wave cannot penetrate this object, the shadow d generated due to the occlusion of this object is distributed along the range direction. According to the radar observation geometry, the length d of the shadow along the range direction and the height H of the ground object relative to the ground have the following relationship(3)d=H∗tan(θ)

Therefore, the length and shape of the shadow are related to the height and shape of the ground object at point B. If we can determine whether there is a prominent elevation ground object on the left-hand side of the low-power area (i.e., the near-range end of the radar in the range direction) and whether the shadow generated by this prominent elevation ground object can match the low-power area, we can distinguish between shadows and water bodies.

Therefore, the first step of the shadow–water separation algorithm is to find prominent elevation ground objects, and the second step is to determine the shape correlation between the shadow and the prominent elevation ground object based on the ratio of the number of corner points.

In order to distinguish between shadows and water bodies in the weak scattering areas, this paper combined elevation and image corners for classification. The radar observation geometry is a typical oblique-view imaging geometry. If there are tall ground objects, such as buildings and vegetation, blocking in the scene, shadows related to the shapes of these tall ground objects will be formed in the direction of the radar line of sight. As shown in Figure 3, the radar observation geometry and the elevation in the scene are utilized to subdivide the weak scattering areas.

Extract the prominent elevation areas. Firstly, calculate the average elevation within each grid based on the grids obtained from the polarimetric SAR image segmentation and the digital elevation model. Then, compare the average elevations of adjacent grids in the order from the near-range end to the far-range end of the radar. If the average elevation of the grid at the near-range end is higher than that of the grid at the far-range end, mark the grid at the near-range end as a prominent elevation area. As shown in Figure 4, the left side of the indicated area is the near-range end, and the right side is the far-range end. The numbers within the grids represent the average elevations, and the grids marked in red are the prominent elevation areas, whose average elevations are all higher than those of the adjacent grids on their right ends.Identify suspicious shadow areas. For each weak scattering area, check if there is a prominent elevation area adjacent to its near end. If such an area exists, the weak scattering area is considered a suspicious shadow area; otherwise, it is classified as a water body area.Confirmation of Shadow Areas. Conduct the confirmation of shadow areas by comparing the boundaries of the suspected shadow areas with those of the prominent elevation areas at their near-range ends. If the suspected shadow areas are caused by the obstruction of the prominent elevation areas at their near-range ends, then their boundaries will have a high degree of similarity. This similarity is quantified by the number of corner points. A simple corner point extraction method is presented below.

As illustrated in Figure 5, initially, in accordance with the segmentation outcomes of the polarimetric SAR image, the gray level of the pixels within the area for corner point extraction is set to 1, while that of the pixels outside this area is set to 0. Subsequently, an N × N template is constructed, with N being an odd number. The center of this template is then positioned on a particular pixel located at the edge of the area where the corner points are to be extracted. The sum of the gray values within the template is computed. If this sum is less than N × N/3, the edge point where the center of the template resides is recorded as a corner point. By sliding this template and repeating the aforesaid process, all the corner points can be effectively extracted.

The above-mentioned method is used to extract the number of corner points of the suspected shadow area and the prominent elevation area at its near-range end. Suppose the number of corner points extracted from the suspected shadow area is $n_{1}$ and the number of corner points of the adjacent prominent elevation area at its near-range end is $n_{2}$. If Equation (4) holds; it indicates that there is not much difference in the corner point data between the two areas. It is considered that the suspected shadow area is caused by the obstruction of the prominent elevation area, so it is classified as a shadow area; otherwise, it is classified as a water body.(4)$$\frac{\left|n_{1}-n_{2}\right|}{max(n_{1},n_{2})}<0.2$$

### 2.3. The Classification of Strong Scattering Areas Using Polarimetric Information

For strong scattering areas, since the signal-to-noise ratio is high enough, the utilization of polarimetric information becomes particularly crucial. Through model-based target decomposition [13], we can extract the powers of different scattering components from pixels, including odd-bounce scattering components, even-bounce scattering components, and volume scattering components. These scattering components provide important bases for land cover classification.

Polarimetric Information Extraction. Firstly, the preprocessed polarimetric synthetic aperture radar data, namely, the data of the four polarization channels HH, HV, VH, and VV, is utilized to extract the polarimetric information of the strong scattering areas. This information is obtained by constructing the coherency matrix $T_{3}$, as shown in Equation (5). The coherency matrix is fundamental in the analysis of polarimetric SAR data, as it contains all the second-order statistical information of the electromagnetic wave scattering within the pixels.(5)$$T_{3}=\langle kk^{*T}\rangle$$(6)$$k=\frac{1}{\sqrt{2}}{\left[HH_{m}+VV_{m}HH_{m}-VV_{m}2HV_{m}\right]}^{T}$$ where the superscript * represents conjugate, T represents transpose, and 〈.〉 represents the multi-look average.Calculation of Scattering Components. Based on the coherency matrix $T_{3}$, we can calculate the odd-bounce scattering component $P_{s}$, the even-bounce scattering component $P_{d}$, and the volume scattering component $P_{v}$.If Equation (7) holds, then the odd-bounce scattering component $P_{s}$ is dominant and the even-bounce scattering component $P_{d}$ is negligible. The three components are solved through Equation (8).(7)$$T_{3}(1,1)>T_{3}(2,2)+T_{3}(3,3)$$(8)$$\left\{\begin{array}{l} P_{s}=\frac{T_{3}^{2}(1,1)-4T_{3}(1,1)T_{3}(3,3)+T_{3}^{2}(1,2)+4T_{3}^{2}(3,3)}{T_{3}(1,1)-2T_{3}(3,3)}\\ P_{d}=\frac{\left[T_{3}(1,1)-2T_{3}(3,3)]\cdot [T_{3}(2,2)-T_{3}(3,3)]-T_{3}^{2}(1,2\right)}{T_{3}(1,1)-2T_{3}(3,3)}\\ P_{s}=4T_{3}(3,3) \end{array}\right.$$If Equation (9) holds, then the even-bounce scattering component $P_{d}$ is dominant and the odd-bounce scattering component $P_{s}$ is negligible. The three components are solved through Equation (10).(9)$$T_{3}(1,1)\leq T_{3}(2,2)+T_{3}(3,3)$$(10)$$\left\{\begin{array}{l} P_{s}=\frac{\left[T_{3}(1,1)-2T_{3}(3,3)]\cdot [T_{3}(2,2)-T_{3}(3,3)]-T_{3}^{2}(1,2\right)}{T_{3}(2,2)-T_{3}(3,3)}\\ P_{d}=\frac{T_{3}^{2}(1,2)+{\left[T_{3}(2,2)-T_{3}(3,3)\right]}^{2}}{T_{3}(2,2)-T_{3}(3,3)}\\ P_{s}=4T_{3}(3,3) \end{array}\right.$$Land Cover Classification. After obtaining the scattering components of each pixel, we can conduct land cover classification based on the relative magnitudes of these components. Different land cover types usually have different scattering characteristics, which play a decisive role in land cover classification.

For vegetation areas, due to the presence of anisotropic branches and leaves in these areas, which is modeled as volume scattering, there is relatively strong volume scattering in this region. Additionally, since the trunks and the ground form a dihedral angle, there is a small amount of even-bounce scattering in relatively sparse vegetation areas; for relatively sparse and low-lying vegetation areas, as there is some bare soil in the area, there is also a small amount of odd-bounce scattering. Therefore, in vegetation areas, the volume scattering content is the largest, followed by the even-bounce scattering or odd-bounce scattering content. The possible power magnitudes of the various scattering components in these areas are as follows: volume scattering $P_{v}$ > even-bounce scattering $P_{d}$ > odd-bounce scattering $P_{s}$ or volume scattering $P_{v}$ > odd-bounce scattering $P_{s}$ > even-bounce scattering $P_{d}$.

Building areas are mainly composed of man-made buildings such as houses. In these areas, due to the dihedral angle formed by the walls and the ground, relatively strong even-bounce scattering may occur. Additionally, since areas such as rooftops are relatively flat, a large amount of odd-bounce scattering components may be generated. Therefore, in man-made building areas, the content of even-bounce scattering or surface scattering is the largest, and even-bounce scattering is the specific scattering component of these areas. The possible power magnitudes of the various scattering components in these areas are as follows: even-bounce scattering $P_{d}$ > volume scattering $P_{v}$ > odd-bounce scattering $P_{s}$, even-bounce scattering $P_{d}$ > odd-bounce scattering $P_{s}$ > volume scattering $P_{v}$, or odd-bounce scattering $P_{s}$ > even-bounce scattering $P_{d}$ > volume scattering $P_{v}$.

Bare soil areas are mainly composed of bare ground as well as areas such as roads and squares. Mainly odd-bounce scattering is generated, and there is not a large amount of even-bounce scattering. However, due to the Bragg scattering produced by the bare soil or the presence of a small amount of weeds, there may be some volume scattering. The specific scattering component of these areas is odd-bounce scattering. Therefore, the possible power magnitudes of the various scattering components in bare soil areas are as follows: odd-bounce scattering $P_{s}$ > volume scattering $P_{v}$ > even-bounce scattering $P_{d}$.

The next step is to obtain the odd-order scattering power $P_{s}$, even-order scattering power $P_{d}$, and volume scattering power $P_{v}$ for each pixel, and sort them in descending order.

Based on the sorted results of the power of each scattering component, the pixels are classified according to the following rules:

If $P_{v}$ > $P_{s}$ > $P_{d}$, classify the pixel as vegetated;

If $P_{v}$ > $P_{d}$ > $P_{s}$, classify the pixel as vegetated;

If $P_{s}$ > $P_{v}$ > $P_{d}$, classify the pixel as bare soil;

If $P_{s}$ > $P_{d}$ > $P_{v}$, classify the pixel as built-up;

If $P_{d}$ > $P_{s}$ > $P_{v}$, classify the pixel as built-up;

If $P_{d}$ > $P_{v}$ > $P_{s}$, classify the pixel as built-up.

According to the above rules, the strong scattering areas are classified into bare soil, building, and vegetation areas.

### 2.4. Fine Classification of Vegetation Areas

Fine classification of the vegetation areas is carried out through vegetation height and leaf type. The DEM of vegetation can be obtained through interferometric information. However, the DEM does not represent the height of the vegetation itself. It is necessary to obtain the vegetation ground elevation that can characterize the height of the vegetation itself so as to distinguish vegetation of different heights. The anisotropy parameter was first proposed by Neumann [7,14]. It can characterize the transformation of the shape of scattering particles from spherical to dipolar in the polarization plane, and it can distinguish vegetation with different leaf types, classifying the vegetation into coniferous vegetation and broad-leaved vegetation.

#### 2.4.1. Vegetation Ground Elevation and Classification of Tall and Short Vegetation

The DEM represents the elevation of ground objects relative to the same reference plane, such as the sea level. It cannot directly reflect the actual height of the ground objects themselves, for example, the height of vegetation or the height of buildings. In the actual process of land cover classification, the height of ground objects is a key distinguishing factor. To classify land cover more accurately, the actual height information of ground objects needs to be considered. In this paper, the actual height of ground objects was extracted through the results of polarization classification and the DEM.

Extract the DEM and set the DEM of the weak scattering areas to zero. Because the power in the weak scattering areas is extremely weak, the extracted DEM is not accurate.Extract the actual height of the ground objects, that is, the height of the ground objects relative to the ground, by combining the classification results of the strong scattering areas in Section 2.3. Since the physical scattering form of the ground is generally odd-bounce scattering, through the classification in Section 2.3, classify the land cover type of the ground as bare soil. Obtain the elevation of the vegetation relative to the nearby bare soil by subtracting the elevation of the nearby bare soil from the elevation of the vegetation, which is the vegetation-to-ground elevation. The specific process of this step is shown in Figure 6.

Subtract the average DEM of the nearest bare soil from the DEM of each vegetation pixel to obtain the elevation of the ground object relative to the ground. Let the vegetation pixel be $p_{i}$, where i∈{1,…,N}, N is the total number of pixels, $DEM_{pi}$ represents the DEM elevation of pixel $p_{i}$, gij represents the *jth* bare soil pixel within the rectangular window centered on pixel $p_{i}$, where o<j<W×W, and W is the window size. $DEM_{gij}$ represents the DEM elevation of the bare soil pixel gij and $H_{pi}$ represents the elevation of pixel pi relative to the ground. Then, the DEM elevation $H_{pi}$ of pixel *pi* relative to the ground is shown in Equation (11).(11)$$H_{pi}=DEM_{pi}-\frac{\sum_{j=1}^{W\times W}DEMgij}{W\times W}$$

For each pixel $p_{i}$, search for the bare soil pixels within a rectangular window with a window size of W centered on this pixel and calculate the average DEM of the bare soil pixels gij within this rectangular window. Then, the elevation $H_{pi}$ of the central pixel $p_{i}$ relative to the ground is the elevation $DEM_{pi}$ of this pixel minus the average DEM of the bare soil pixels gij within the rectangular window.

3.Classify the vegetation into two types, namely, low and high vegetation, based on the elevation data relative to the ground. The elevation of the vegetation relative to the ground obtained in the above steps is approximately regarded as the actual height of the vegetation. Therefore, by setting a threshold value T1 for the elevation relative to the ground, the vegetation can be classified into two types: low and high vegetation. The threshold value can be determined by the method of maximum inter-class variance of the histogram or by using the method of support vector machine through sample training.If $H_{pi}\leq T1$, it is low vegetation;If $H_{pi}\geq T1$, it is high vegetation.

#### 2.4.2. Extraction of Anisotropy Parameters and Classification of Vegetation Leaf Types

The leaf types of vegetation can be distinguished by anisotropy parameters, which were first proposed by Neumann [7,13]. These parameters can characterize the transformation of scatterer shape from spherical to dipolar in the polarization plane. The anisotropy parameters can also represent the transformation of vegetation from broad-leaved to needle-leaved. The modulus of the anisotropy parameters $\left|\delta \right|$ is obtained using Equation (12).(12)$$\left|\delta \right|=\sqrt{\frac{\langle |S_{HH}-S_{VV}{|}^{2}+4<|S_{HV}{|}^{2}\rangle }{\langle |S_{HH}+S_{VV}{|}^{2}\rangle }}$$
where $\langle .\rangle$ represents the operation of taking the average of multiple looks.

According to the modulus of anisotropy parameters decomposed by Neumann, the vegetation is divided into coniferous and broad-leaved vegetation. First, a threshold *T* is determined, which is determined through the method of maximum inter-class variance in histograms or through sample training using support vector machines. Then, the vegetation is classified into coniferous and broad-leaved vegetation based on the modulus of anisotropy parameters and the threshold *T*.

The selection of the threshold value *T* is undoubtedly a challenging task. Its value is often closely related to diverse observation systems, different observation angles, and a wide variety of vegetation types. Given these circumstances, it is almost impossible to directly provide a universal threshold value that can be used immediately. Generally, it is necessary to rely on a large amount of experimental data and use traditional algorithms, such as decision trees and support vector machines, or machine learning algorithms to accurately determine a suitable threshold value. Here, we assumed that the threshold value *T* has been successfully obtained through the above effective methods.

Specifically, if the modulus of anisotropy parameters is greater than the threshold *T*, it is considered as coniferous vegetation; otherwise, it is considered as broad-leaved vegetation.

By combining the terrain height of vegetation and anisotropy parameters, the fine classification of vegetation is achieved. As shown in Figure 7, if the terrain height of vegetation is less than threshold *T*1 and the modulus of anisotropy parameters is less than threshold *T*, the vegetation pixel is identified as broad-leaved low vegetation; if the terrain height of vegetation is less than threshold *T*1 and the modulus of anisotropy parameters is greater than threshold *T*, the vegetation pixel is identified as needle-leaved low vegetation; if the terrain height of vegetation is greater than threshold *T*1 and the modulus of anisotropy parameters is less than threshold *T*, the vegetation pixel is identified as broad-leaved tall vegetation; and if the terrain height of vegetation is greater than threshold *T*1 and the modulus of anisotropy parameters is greater than threshold *T*, the vegetation pixel is identified as needle-leaved tall vegetation.

## 3. Results

The PolInSAR data provided by the N-SAR system [15,16] of Nanjing Institute of Electronic Technology is used to validate the method in this paper. The data were collected in the Weihe River area of Shaanxi Province at X-band. These data have already undergone data preprocessing, DEM acquisition, and polarimetric SAR image segmentation. Figure 8 shows the Pauli fusion result of the data, Figure 9 shows the DEM map, and Figure 10 shows the polarimetric SAR image segmentation result.

### 3.1. Classification Results of Strong and Weak Scattering Areas

Weak scattering areas are mainly water bodies and shadow areas. Under the X-band, the penetration effect of electromagnetic waves on water bodies is limited. Most of the electromagnetic wave energy is specularly reflected on the water surface, and the signals received by the radar are extremely weak. For shadow areas, due to the obstruction of ground objects, the echoes in this area are mainly radar system noise and sidelobe interference from other areas.

The classification results of the strong and weak scattering areas of the experimental data are shown in Figure 11, where the black areas are the weak scattering areas and the yellow areas are the strong scattering areas.

### 3.2. Classification Results of Weak Scattering Areas and Strong Scattering Areas

Figure 12 displays the classification results of weak scattering areas in the experimental data, where the black areas represent shadows and the blue areas represent water bodies. By this method, we can effectively distinguish between shadows and water bodies in weak scattering areas, providing accurate information for subsequent land cover classification.

Figure 13 shows the classification results of strong scattering areas based on the method in this paper. As is seen from the figure, the proposed method can effectively distinguish between bare soil, buildings, and vegetation. The red areas represent building areas, the green areas represent vegetation areas, the yellow areas represent bare soil, and the black areas represent weak scattering areas.

### 3.3. Classification Results of Vegetation Areas

As shown in Figure 9a, the DEM of the Weihe Bridge area was extracted. It can be seen that the river area and shadow areas have extremely low power, so the interferometric information is invalid, and the obtained elevation information is relatively chaotic. Therefore, the elevation of the weak scattering areas was not used in subsequent processing.

As shown in Figure 9b, the ground elevation of the Weihe Bridge area was generated. By comparing Figure 9a with Figure 9b, it can be found that the elevations of the building and vegetation areas in Figure 8 are highlighted, removing the influence of terrain and reflecting the true height information of real objects.

The results of the refined vegetation classification, which utilizes the elevation of vegetation relative to the ground, are presented in Figure 14. The method proposed in this paper is capable of accurately subdividing vegetation while maintaining the details of the image. Table 1 shows the confusion matrix of this method and also reports the calculated recall and precision values.

Judging from the data results, the proposed method demonstrates outstanding performance. Its overall classification accuracy reaches an impressive 90.2%, with a Kappa coefficient of 0.876, and the recall values are all above 70%. Specifically, the precision values of bare soil, water bodies, buildings, short broad-leaved vegetation, and tall broad-leaved vegetation are generally high, roughly around 80%.

For comparison, we present the classification results based on the support vector machine (SVM). The kernel used in the SVM is the Gaussian kernel, and the multi-class discrimination scheme is the One-Versus-Rest (OVR) method.

As shown in the right part of Figure 15, the bare soil is misclassified as shrubs. In the middle-left part and the lower-right part of Figure 15, there are also many areas where the bare soil is misclassified as artificial buildings. Additionally, some grasslands are misclassified as trees, and a considerable portion of the shrub areas are not correctly identified. There are also a large number of misclassifications in the shadow areas and water bodies. Table 2 shows the classification results obtained by the support vector machine (SVM) method and also presents the calculated recall and precision values.

An in-depth analysis of the classification results reveals that when the support vector machine (SVM) is employed for object classification of PolInSAR data in the Weihe Bridge, the overall classification accuracy is merely 76.8%, and the Kappa coefficient stands at 0.709.

## 4. Discussion

This paper proposes a new polarimetric interferometric SAR land cover classification method. The core of this method lies in first dividing the area into strong scattering areas and weak scattering areas using power information and then adopting different classification strategies for these two types of areas. This process of subdividing from large categories to small categories significantly improves classification accuracy.

In addition, this paper also introduces the concept of the elevation of ground objects relative to the ground. Traditional interferometric SAR can only provide the elevation information of ground objects relative to the sea level, but it cannot reflect the actual height of ground objects. For this reason, by combining DEM data and polarimetric classification results, this paper successfully extracts parameters that can reflect the actual height of ground objects. This parameter is of great value in the inversion and classification of ground object elevation.

The classification results of the measured data show that the method proposed in this paper performs excellently in terms of both classification types and classification accuracy. In particular, the accuracy of bare soil, water bodies, buildings, and tall broad-leaved vegetation even exceeds 90%. While fully preserving the details of the image, this method successfully achieves a high classification accuracy, which fully demonstrates its remarkable superiority. It provides a brand-new perspective and method for land cover classification and has broad application prospects.

In contrast, other classification methods have obvious deficiencies. For example, when using the support vector machine (SVM) to classify PolInSAR data, the precision and recall of shadows and water bodies are at a relatively low level. This clearly indicates that this method has significant defects in accurately identifying shadows and water bodies. In addition, the recall of shrubs and trees is even below 50%, which undoubtedly shows that this classification method has difficulty effectively distinguishing between shrubs and trees.

The classification results of the measured data show that the method in this paper performs excellently in both classification types and classification accuracy, providing a brand-new perspective and method for land cover classification with broad application prospects.

## Figures and Tables

**Figure 1 sensors-25-01996-f001:**
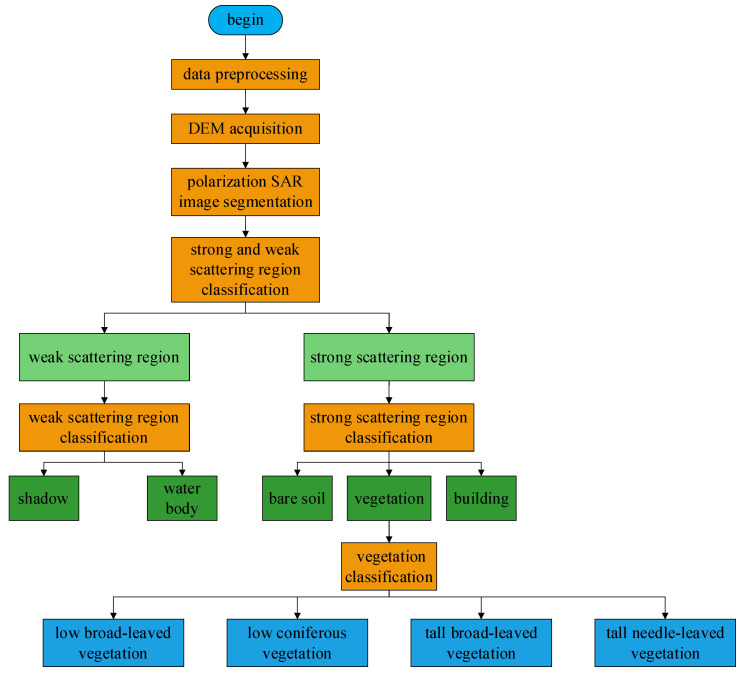
The main flowchart of the method in this paper.

**Figure 2 sensors-25-01996-f002:**
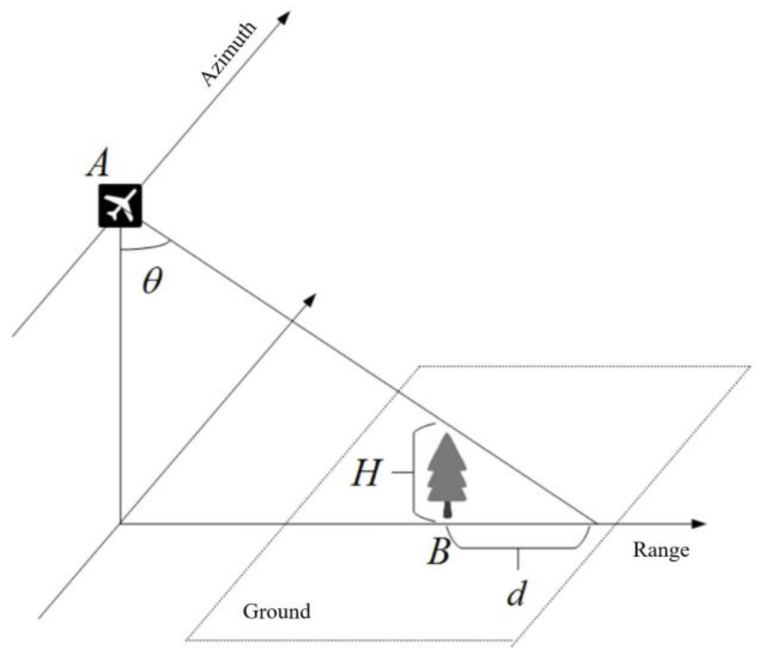
Schematic diagram of shadow formation in SAR images.

**Figure 3 sensors-25-01996-f003:**
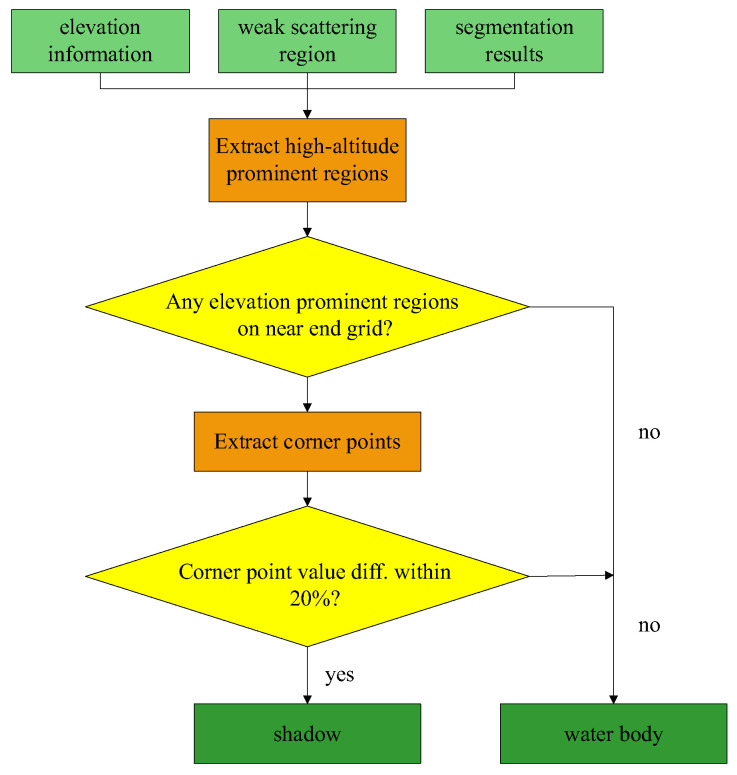
Classification process for weak scattering areas.

**Figure 4 sensors-25-01996-f004:**
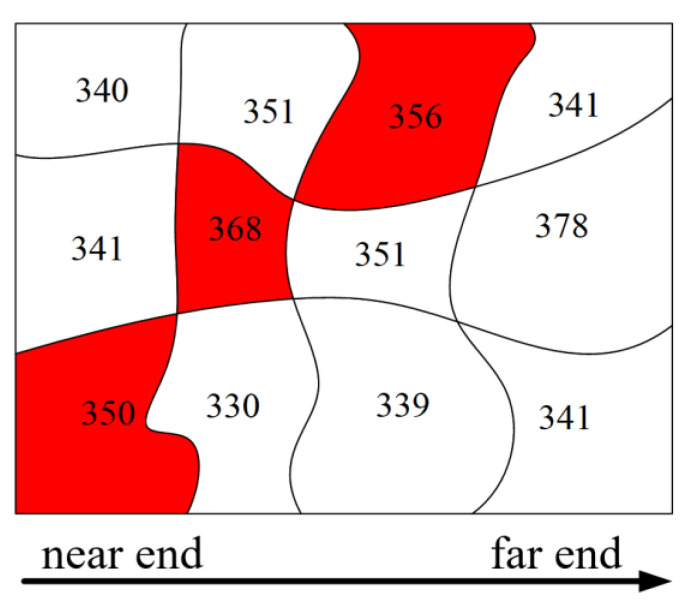
Schematic diagram of the prominent elevation area.

**Figure 5 sensors-25-01996-f005:**
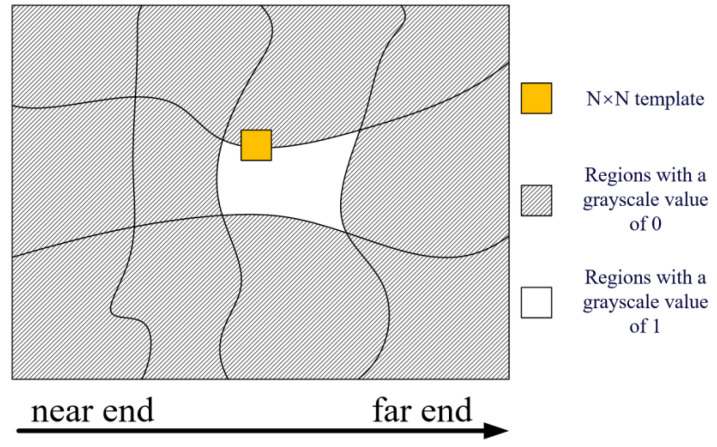
Diagram of corner point extraction.

**Figure 6 sensors-25-01996-f006:**
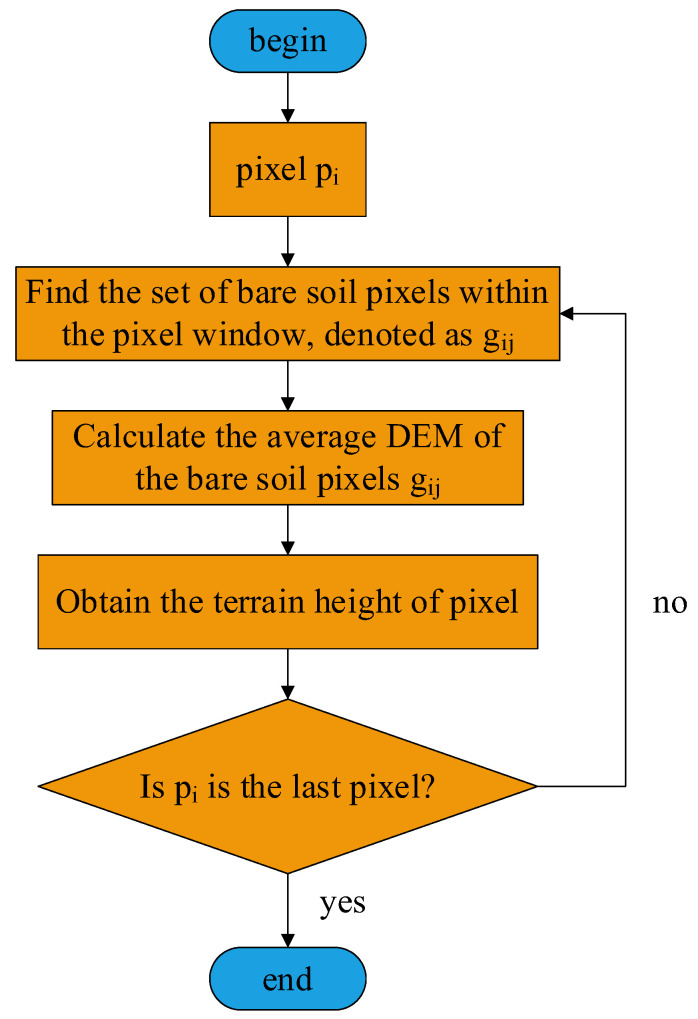
Flow chart of vegetation elevation extraction.

**Figure 7 sensors-25-01996-f007:**
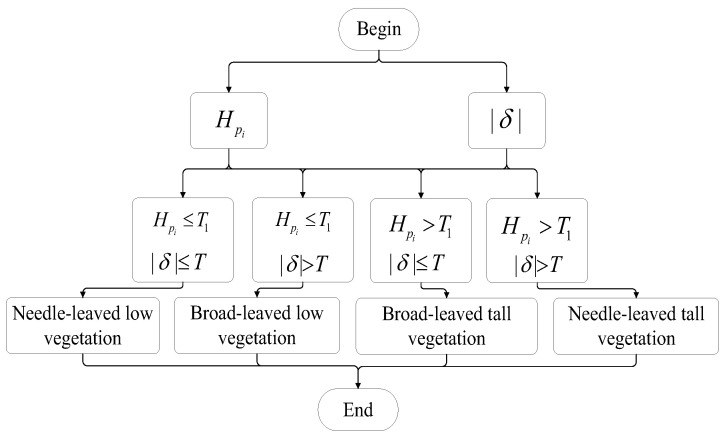
Flowchart for vegetation classification.

**Figure 8 sensors-25-01996-f008:**
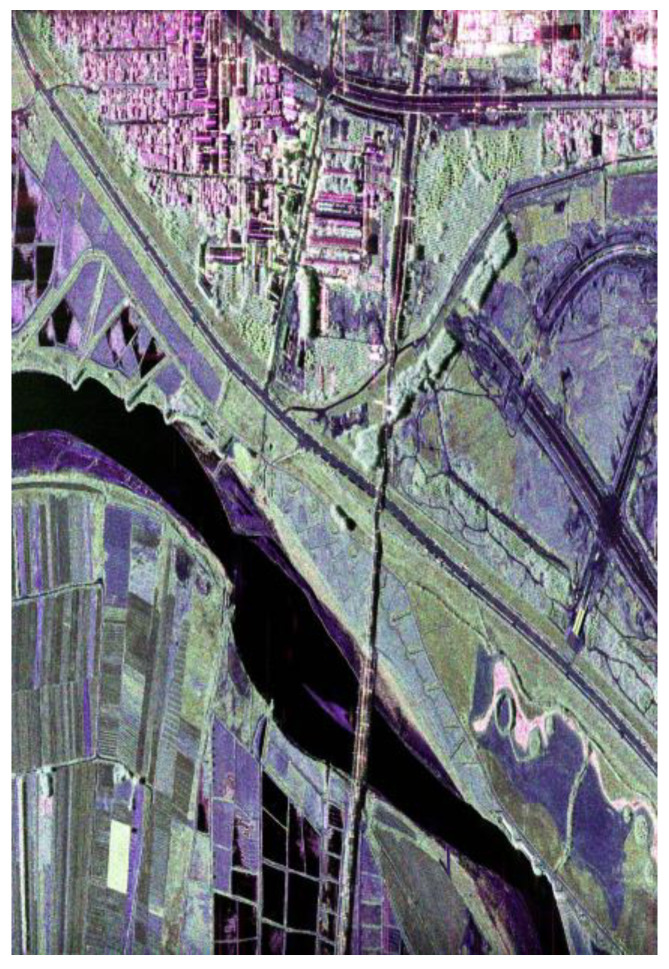
The Pauli fusion result of the data.

**Figure 9 sensors-25-01996-f009:**
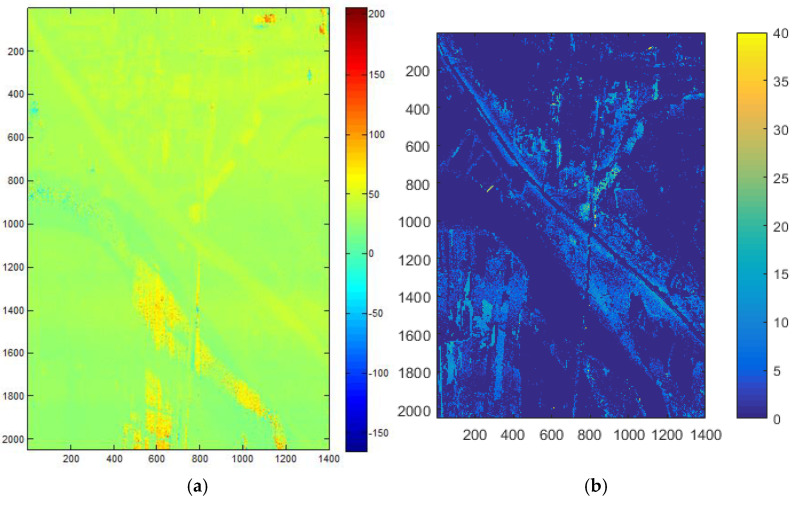
(**a**). The original Digital Elevation Model (DEM) map. (**b**). The vegetation ground elevation map.

**Figure 10 sensors-25-01996-f010:**
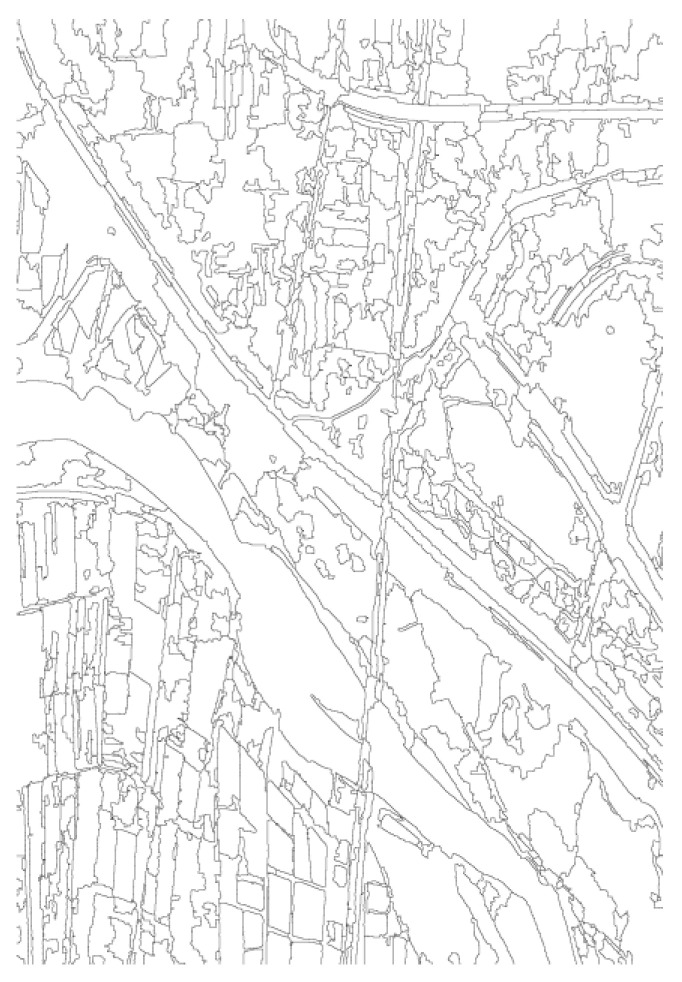
The polarimetric SAR image segmentation result.

**Figure 11 sensors-25-01996-f011:**
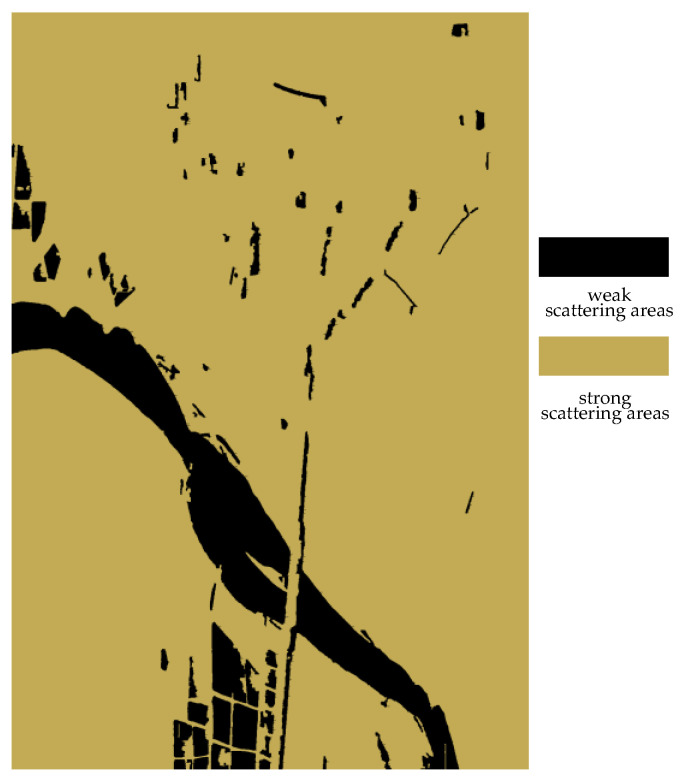
The classification results of the strong and weak scattering areas of the experimental data.

**Figure 12 sensors-25-01996-f012:**
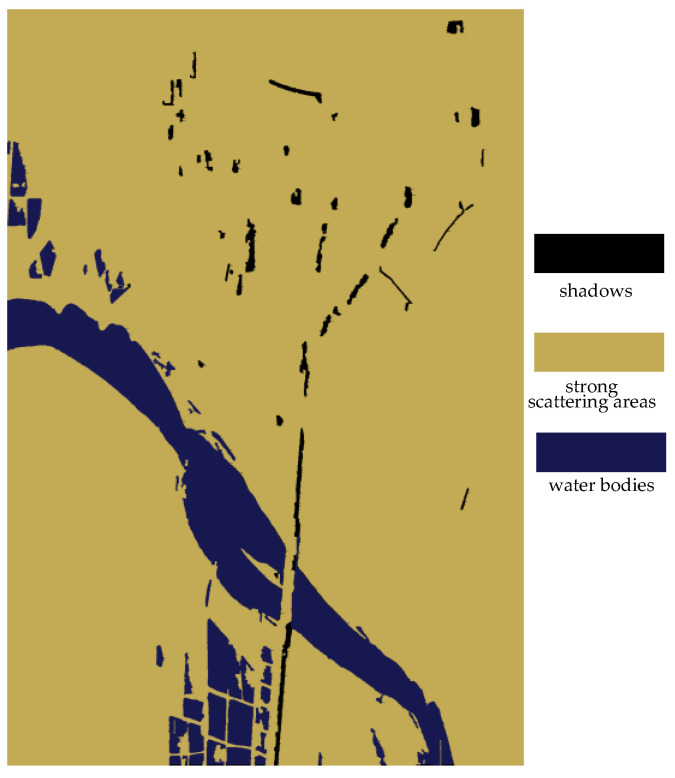
The classification results of weak scattering areas.

**Figure 13 sensors-25-01996-f013:**
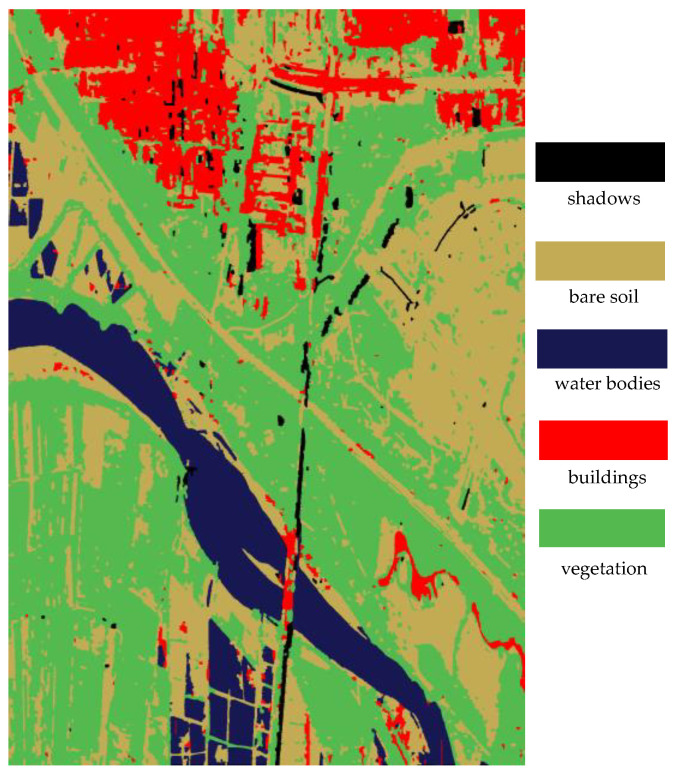
The classification results of strong scattering areas.

**Figure 14 sensors-25-01996-f014:**
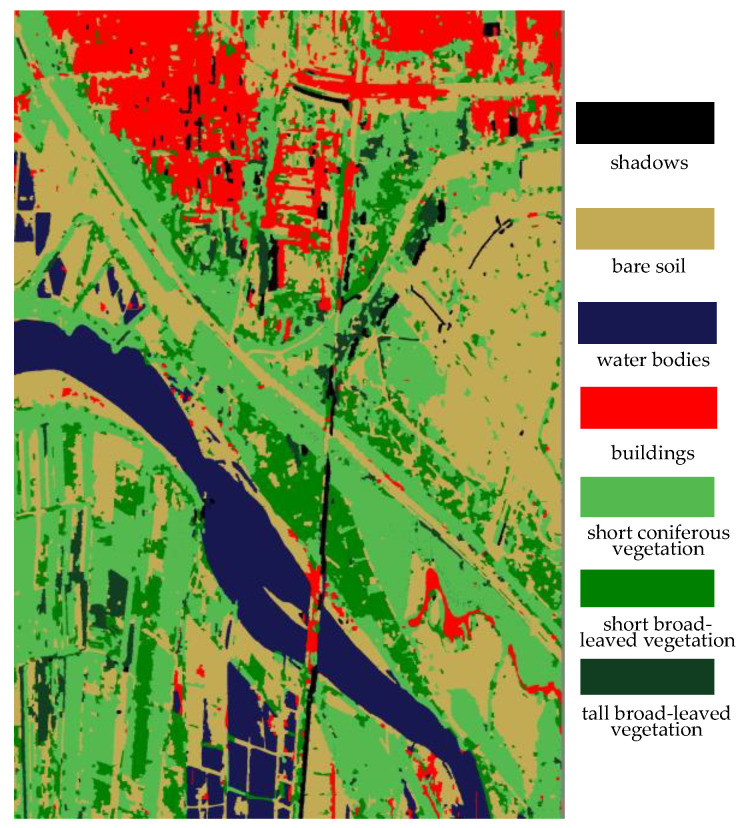
The result of the refined classification of vegetation.

**Figure 15 sensors-25-01996-f015:**
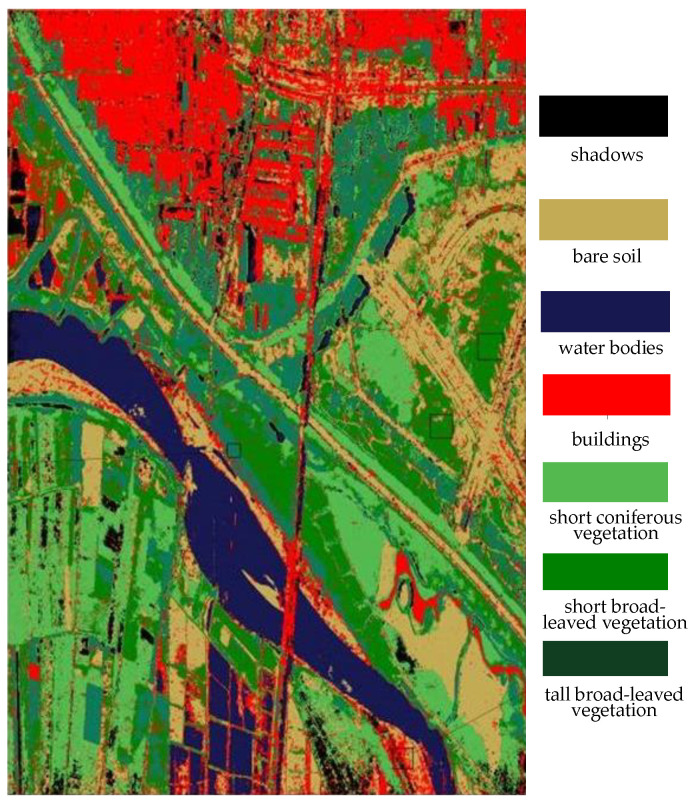
The classification results based on the support vector machine (SVM).

**Table 1 sensors-25-01996-t001:** The classification confusion matrix of the proposed method in this paper for the validation data.

Classification Category	Ground Type							
	Shadow	Bare Soil	Water Body	Building	Short Coniferous Vegetation	Short Broad-Leaved Vegetation	Tall Broad-Leaved Vegetation	Precision %
Shadow	6837	12	1889	409	10	102	18	73.7
Bare soil	433	132,479	1980	8244	1128	740	740	90.9
Water body	0	4	198,683	0	0	0	0	99.9
Building	113	490	334	144,929	0	0	0	99.4
Short coniferous vegetation	958	3351	803	16,653	60,962	6256	4895	64.9
Short broad-leaved vegetation	701	7599	155	5420	1353	55,713	1217	77.2
Tall broad-leaved vegetation	0	58	0	201	0	1386	19,868	92.4
Recall%	75.6	92.0	97.5	82.4	96.1	86.8	74.3	

**Table 2 sensors-25-01996-t002:** The classification confusion matrix of the SVM method for the validation data.

Classification Category	Ground Type							
	Shadow	Bare Soil	Water Body	Building	Short Coniferous Vegetation	Short Broad-Leaved Vegetation	Tall Broad-Leaved Vegetation	Precision %
Shadow	2313	1814	4530	9210	2897	390	984	10.4
Bare soil	552	78,897	5203	2270	1862	450	565	87.9
Water body	4395	98	189,689	927	4	27	215	97.1
Building	1739	12,950	4105	145,749	2039	2523	3162	84.6
Short coniferous vegetation	32	6221	138	3749	40,652	4460	1688	71.4
Short broad-leaved vegetation	2	41,842	38	3680	6677	52,475	2394	49.0
Tall broad-leaved vegetation	9	2171	141	10,271	9312	3872	17,730	40.8
Recall%	25.6	54.8	93.1	82.9	64.1	81.8	66.3	

## Data Availability

The original contributions presented in this study are included in this article; further inquiries can be directed to the corresponding author.

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
