# Peer review of "Innovative Polarimetric Interferometric Synthetic Aperture Radar Land Cover Classification: Integrating Power, Polarimetric, and Interferometric Information for Higher Accuracy"

_sensors, 2025, doi:10.3390/s25071996_

Round 1
Reviewer 1 Report
Comments and Suggestions for Authors
This paper suggests a PolInSAR classification method using power, polarimetric, and interferometric data to improve accuracy. It distinguishes land cover from areas of strong and weak scattering, and weak areas (shadows, water) and strong areas (vegetation, buildings, soil) are distinguished further. Vegetation is characterised by elevation and anisotropy as well. It was implemented on N-SAR data and achieved 90.2% accuracy and a Kappa coefficient of 0.876.
The paper is well-written and clear, but the following concerns should be addressed and clarified:
-On page 3, line 106, substitute the word chapter with paper.
-Combining power, polarimetric, and interferometric data creates greater computational complexity, which might be difficult for big data or real-time applications. Investigate employing more efficient algorithms, e.g., deep learning, to enhance processing speed without compromising accuracy.
- Classification may be affected by environmental conditions, i.e., atmospheric variation or sensor degradation, especially for water and shadow. More environmental information or advanced filtering procedures would be helpful to enhance robustness.
- Using NESZ as a uniform measure of sensitivity overestimates system performance and fails to adequately consider the heterogeneity of backscattering response, with the potential for introducing error in low-backscatter regions.
- Thermal noise is more evident in areas of poor backscatter, e.g., water or flat ground, and can mask weak signals and cause detection and classification errors.
Reviewer 2 Report
Comments and Suggestions for Authors
An original classifier structure combining power, polarimetric and interferometric information is proposed. The goal is to improve classification accuracy. Power information is used to distinguish between areas of strong and weak scattering.
The choice of the recognition threshold is based on the radar range equation, which determines the minimum ratio of the reflected signal power to the receiver's own noise power. This raises questions, since with such separation there is a possibility that pixels containing purely noise information about the surface will fall into the weak scattering region. Such pixels should simply be excluded from consideration. I think this separation criterion is not very successful, this is indirectly confirmed by paragraph 2 on page 5, which introduces a threshold value using a known shadow area as a reference. However, the choice of the shadow area as a reference for use as a reference depends on the quality of the segmentation. If this area is specified by an operator, the classification algorithm is no longer automatic.
The algorithm for separating shadows and bodies of water is heuristic. There are no models for describing these areas using image heights and angles. It mentions a digital terrain model, which is not described in any way. The algorithm is based on comparing average terrain values. Its quality probably depends heavily on the segmentation results. It is unclear how to set decision thresholds in many situations, in particular, when extracting anisotropy parameters and classifying vegetation leaf types.
The classification method is represented by a set of fairly simple procedures that are well known. What is new is the expansion of the class of these procedures by using a wider set of features. However, there is no mathematical or other model describing the combination of these features. Everything is presented on an intuitive and heuristic level. The practical results are presented without comparison with known analogues. The generally accepted characteristics of classifiers, such as Recall-Precision are not used.
Since the characteristics of the classifier significantly depend on the quality of segmentation, the classification task should be considered as a general task, taking into account the quality of segmentation.
Author Response
Thank you very much for taking the time to review this manuscript. Please find the detailed responses in the attachment.

Round 2
Reviewer 1 Report
Comments and Suggestions for Authors
Thanks for addressing my comments.
Author Response
Dear Reviewer,
Thank you very much for your review and positive feedback. We're glad that you think our paper's English is fine, and that the introduction, research design, method description, result presentation, and conclusion - result correspondence all meet your requirements.
We really appreciate your understanding and recognition of our handling of previous comments. This has given us great encouragement. If in the future you have any further suggestions or comments on our paper, please feel free to let us know. We will continue to work hard to improve the quality of our research.
Best regards,
Authors' Team of the Paper: Yifan Xu, Aifang Liu, Youquan Lin, Moqian Wang, Long Huang, Zuzhen Huang
Reviewer 2 Report
Comments and Suggestions for Authors
The problem of combining heterogeneous features in order to improve the quality of classification is very difficult. I welcome your efforts in this direction, and appreciated a number of achievements. I have approved of your work. It seems to me that possible options for combining features should use their interrelationships arising from the physics of processes. I wish you success on this difficult way.
Author Response
Dear Reviewer,
Thank you very much for your recognition and encouragement of our paper. We are honored to learn that you affirm our efforts in combining heterogeneous features to improve the classification quality.
You mentioned that possible options for combining features should utilize the interrelationships arising from the physics of processes. In this study, we indeed fully considered the connections based on physical principles among different types of information (power information, polarimetric information, and interferometric information). For example, when distinguishing between strong and weak scattering areas, we made the division based on the principle of the radar system's response to the backscattering intensities of different ground objects using power information. In the classification of strong scattering areas, based on the scattering characteristics of different ground objects (such as the volume scattering of vegetation and the dihedral angle scattering of buildings, etc., which are physical mechanisms), we extracted scattering components through polarimetric decomposition to achieve classification.
In subsequent research, we will further explore the interrelationships of these features based on physical processes. On the one hand, we will explore more methods for feature extraction and fusion based on physical principles. For instance, we will consider the differences in the interaction of electromagnetic waves with ground objects at different frequencies and introduce new feature parameters. On the other hand, we will attempt to construct a more comprehensive model to describe the relationships among these features, so as to optimize the classification process and further improve the accuracy and reliability of classification.
Thank you again for your valuable suggestions. We will strive to achieve more results in this challenging research direction.
Best regards!
Authors' Team of the Paper: Yifan Xu, Aifang Liu, Youquan Lin, Moqian Wang, Long Huang, Zuzhen Huang